# Multi-Layer Graph Attention Network for Sleep Stage Classification Based on EEG

**DOI:** 10.3390/s22239272

**Published:** 2022-11-28

**Authors:** Qi Wang, Yecai Guo, Yuhui Shen, Shuang Tong, Hongcan Guo

**Affiliations:** School of Electronics and Information Engineering, Nanjing University of Information Science and Technology, Nanjing 210044, China

**Keywords:** graph attention network, gated recurrent unit, node-level and stage-level attention, sleep staging, transitional stage estimator

## Abstract

Graph neural networks have been successfully applied to sleep stage classification, but there are still challenges: (1) How to effectively utilize epoch information of EEG-adjacent channels owing to their different interaction effects. (2) How to extract the most representative features according to confused transitional information in confused stages. (3) How to improve classification accuracy of sleep stages compared with existing models. To address these shortcomings, we propose a multi-layer graph attention network (MGANet). Node-level attention prompts the graph attention convolution and GRU to focus on and differentiate the interaction between channels in the time-frequency domain and the spatial domain, respectively. The multi-head spatial-temporal mechanism balances the channel weights and dynamically adjusts channel features, and a multi-layer graph attention network accurately expresses the spatial sleep information. Moreover, stage-level attention is applied to easily confused sleep stages, which effectively improves the limitations of a graph convolutional network in large-scale graph sleep stages. The experimental results demonstrated classification accuracy; MF1 and Kappa reached 0.825, 0.814, and 0.775 and 0.873, 0.801, and 0.827 for the ISRUC and SHHS datasets, respectively, which showed that MGANet outperformed the state-of-the-art baselines.

## 1. Introduction

Sleeping is a crucial condition to maintain normal cognitive function. Long-term sleep deprivation causes normal cognitive diseases, such as learning and memory disorders [1,2]. Different sleep stages are complex and dynamic biological processes that periodically repeat. Polysomnography (PSG) contains physiological signals such as electroencephalogram (EEG) and electroophthalmogram (EOG) [3]. In a clinical setting, the transition rules between sleep stages assist experts to accurately classify sleep stages within frameworks determined by the American Academy of Sleep Medicine (AASM) and the Rechtschaffen and Kales (R&K) standards [4,5], which divide human sleep stages into five classes: wake (W), rapid eye movement (REM), non-REM1 (N1), non-REM2 (N2), and non-REM3 (N3) [6,7].

Automatic sleep staging has wide practical value in clinical medicine. Researchers have used traditional machine learning algorithms based on time and frequency domain features, random forest (RF) [8], and support vector machines (SVM) [9] for sleep stage classification. In recent years, with the development of deep learning, it has been widely used to classify sleep stages with its powerful representational ability. [10,11,12,13]. However, there are some disadvantages to the realization of sleep staging using convolutional neural networks [14]. Generally speaking, there are three types of information fusion in EEG signals processing: spatial information fusion, temporal information fusion, and spatial and temporal information fusion [15]. It not only fuses the relationship dynamics of EEG signals in different brain regions, such as correlation, but also calculates time-frequency domain features at different times through fusion. Traditional CNN architecture and 2D time-frequency signal processing methods could well process the temporal information fusion of EEG signals. Unfortunately, the fixed convolution kernel cannot maintain the translation invariance of non-Euclidean distance signals or directly capture effective node information. Therefore, it cannot fuse and process rich spatial information. Jia et al. [16] considered using the graph structure to establish the functional connectivity and physical proximity of EEG brain regions.

Furthermore, brain regions are located in non-Euclidean space, and different brain regions control various functions of the human body [17]. For example, the brainstem has the function of maintaining circulation and respiration, and the telencephalon dominates the activities of the higher cortex, such as smell, hearing, and other systemic sensations, as well as body movement. Research has determined that the whole of the human brain function structure is strongly influenced by regional anatomical relationships, and the degree of interaction between regions varies [18]. When updating the current node representation of GCN, all adjacent nodes in the brain region will be randomly assigned static important weights. As a result, neighboring nodes closely related to the current node functions are randomly assigned larger or smaller weights, causing the importance of features of node functions to be incorrectly amplified or ignored [19].

Inspired by these recent works, we propose a multi-layer graph attention sleep stage classification network (MGANet) to distinguish sleep stages in the graph structure; the framework is shown in Figure 1. During training, the influence weights of neighbor channels in the EEG spatial domain and temporal domain are calculated, and the constructed graph structure can reflect the different interaction of functional connectivity between brain regions. Multilayer GATs use stacked layered space-time characteristics to capture information, and reduce the graph structure model parameters. This model successfully applies node-level attention and stage-level attention to sleep classification, dynamically establishing the interaction between channels, and combining RNN and GAT to extract the most representational spatial and temporal features.

In general, the sleep staging model based on a multi-layer graph attention network has the following characteristics:

(1) A node-level attention mechanism is applied to effectively calculate and dynamically update the importance of adjacent channels to the current channel, while considering the spatial location and functional connections of brain regions. (2) The parallel operation of multi-layer graph attention convolution of MGANet improves the computational efficiency of the model. When calculating output features and executing attention operations, all nodes and adjacent edges run in parallel. (3) A transition stage estimator is constructed to accurately classify sleep stages by expressing the features of the transition stage epoch through the stage-level attention vector. Experimental results show that MGANet achieves state-of-the-art performance in sleep staging. The remaining sections of this paper are organized as follows. Section 2 introduces the related work. Section 3 describes the materials and our method. Section 4 outlines the results and discussions, and Section 5 summarizes future work.

## 2. Related Work

In early studies, manual sleep stage classification by human sleep experts was tedious and time-consuming, with highly subjective classification results [20,21]. The application of traditional sleep staging methods includes the use of machine learning, such as RF and SVM. Tsinalis et al. [13] used the method of manual extraction of sleep spectrum features, and used stacked sparse autoencoders for integrated learning. Such methods require manual features extraction and a large amount of prior knowledge, and are gradually replaced by deep learning algorithms that automatically extract features.

Subsequently, a single-channel sleep staging model with the bidirectional long short term memory (BiLSTM) network was designed by Supratak et al. [10]. Later, Chambon et al. [11] used a hybrid neural network to extract time-invariant features and time-contextual features. Phan et al. [22] proposed a multi-variable and multi-modal algorithm, which automatically extracted sleep features by using multi-modal physiological signals and auxiliary classification of sleep rules. Chriskos et al. [23] processed sleep stage staging tasks as sequence-to-sequence tasks, and Jia et al. [24] established a sleep staging framework based on CNN to extract synchronous features of cerebral cortex interaction. Eldele et al. [25] proposed a model which consisted of feature extraction modules based on a multi-resolution convolutional neural network (MRCNN) and adaptive feature recalibration (AFR). A method of utilizing deep residual networks in mixed queue environments was proposed by Alexander et al. [26].

In addition, graph convolutional networks inherited from CNN are mainly divided into graph convolutional networks based on spectral structure and graph convolutional networks based on spatial structure [19]. Jia et al. [16] used graph structure to represent brain spatial connections for the first time, which adaptively learns the internal connections between different channels represented by an adjacency matrix. Subsequently, they proposed a multi-view spatial-temporal graph convolutional network with domain generalization, which constructed two brain views by utilizing functional connectivity and the physical proximity of brain regions [27].

In this paper, we propose a new general framework, which represents and fuses the deep semantic features of the input EEG data, and effectively learns the significant features and temporal dependence across epochs. The experimental results show that MGANet is superior to the most advanced baseline. In future work, it could be expanded to emotion recognition, speech processing, non-stationary signal analysis, and other fields [28,29,30].

## 3. Materials and Methods

MGANet constructs a new graph structure to describe the brain’s spatial position and temporal contextual information, the architecture of which is shown in Figure 1. We introduce the networks used to construct sleep graph attention networks, and directly outline their theoretical and practical benefits, as well as limitations, compared to previous work.

### 3.1. Preliminaries

Sleep stage. Epochs were divided into five classes after 30 s.

Graph attentional sleep network. This is defined as an undirected graph $G_{A}=(N,C,A)$, where N,C,A denotes vertex, adjacent edge, and the adjacency matrix in GATs, respectively, mapping the brain as an electrode channel, channel connection, and adjacent relationship between channels, respectively. The EEG channel numbers are defined as |N|=n, and the input sequence is defined as ${\vec{S}}_{m+1,N}^{Nc}=\{i_{m+1,1}^{N_{c}},i_{m+1,2}^{N_{c}},\ldots ,i_{m+1,n}^{N_{c}}\}$, where $i_{m+1,n}^{N_{c}}$ represents the center node $N_{c}’s$ spatial features of the $n^{th}$ channel within the m+1 epoch. The node matrix is updated as ${\vec{f}}_{S}=({\vec{f}}_{1}^{\prime },{\vec{f}}_{2}^{\prime },\cdots ,{\vec{f}}_{m}^{\prime }{)}^{T}$ according to graph attention convolution.

Recurrent neural sleep network. The RNN’s sleep time domain feature sequence is defined as ${\vec{S}}_{t+1,N}^{N_{c}}=\{i_{t+1,1}^{N_{c}},i_{t+1,2}^{N_{c}},\ldots ,i_{t+1,n}^{N_{c}}\}$. $i_{t+1,n}^{N_{c}}$ represents $N_{c}’s$ temporal features of the $n^{th}$ channel at t+1 point. The time-frequency features output are finally updated as ${\vec{f}}_{T}=({\vec{f}}_{1}^{\prime },{\vec{f}}_{2}^{\prime },\cdots ,{\vec{f}}_{t}^{\prime }{)}^{T}$.

### 3.2. Spatial Node-Level Graph Attention Sleep Network

#### 3.2.1. Multi-Head Graph Attention Learning

Firstly, we construct the human brain as a graph, with electrodes corresponding to the nodes of the graph and the connections between channels corresponding to the edges of the graph. In addition, the sleep features of each node are taken as its dynamic features, and appropriate weights are assigned to adjacent nodes, corresponding to the influence level of neighbors on the current channel.

In order to adaptively capture spatial location information through brain position nodes, the input is a node features sequence ${\vec{f}}_{S}=({\vec{f}}_{1}^{\prime },{\vec{f}}_{2}^{\prime },\cdots ,{\vec{f}}_{m}^{\prime }{)}^{T}.$ MGANet uses parallel attention operation to assign importance weightings to different channels in the same brain region according to the influence of the interaction between nodes, using the spatial-temporal features of adjacent channels. Specifically, a fixed number of neighborhood edges are selected around the vertex of each electrode, and the learnable weight coefficient matrix $W_{G}$ of sleep space features is obtained. The attention coefficient u, i.e., the importance of neighbor nodes $N_{k},k=1,2,\cdots ,m$ to the center node $N_{c}$, which is calculated by the attention factor a and $W^{G}$ through the shared attention mechanism, and m is the first-order adjacent channel of $N_{c}$.
(1)$$u_{N_{c},N_{k}}=a(W_{N_{c}}^{G}{\vec{f}}_{N_{c}},W_{N_{k}}^{G}{\vec{f}}_{N_{k}})$$

To make the coefficients across different nodes easily comparable, softmax normalization is used to leverage masked self-attention vector α of $N_{c}$ and its neighborhoods’ $N_{k}$ can be obtained through softmax normalization operation, as follows:(2)$$\alpha_{N_{c},N_{k}}={\mathrm{softmax}}_{N_{c}}(\mu_{N_{c},N_{k}})=\frac{\exp(\mu_{N_{c},N_{k}})}{\sum_{l\in n}\exp(\mu_{N_{c},N_{k}})}$$

The LeakyReLU nonlinear function with negative slope β=0.2 is applied to activate the attention vector $\alpha_{N_{c},N_{k}}$ as follows:(3)$$\alpha_{N_{c},N_{k}}=\frac{\exp(\mathrm{LeakyReLU}({\vec{a}}^{T}[W_{N_{c}}^{G}{\vec{f}}_{N_{c}}||W_{N_{k}}^{G}{\vec{f}}_{N_{k}}]))}{\sum_{l\in n}\exp(\mathrm{LeakyReLU}({\vec{a}}^{T}[W_{N_{c}}^{G}{\vec{f}}_{N_{c}}||W_{N_{k}}^{G}{\vec{f}}_{N_{k}}]))}$$
where ${\vec{a}}^{T}$ is the attention mechanism network transpose operation, $W{\vec{f}}_{N_{c}}||W{\vec{f}}_{N_{k}}$ is the feature aggregation to be given weighting. In order to allocate the importance of different sleep features contained in each channel to adjacent nodes, the multi-node attention mechanism is introduced, and the weighted sum of multiple features contained in nodes $N_{c}$ is then averaged, and the final spatial features ${\vec{f}}_{s}$ are obtained after $L^{th}$ attention. The calculation process is shown as follows:(4)$${\vec{f}}_{s}=\sigma \left(\frac{1}{L}\sum_{l=1}^{L}\sum_{N_{k}\in m}\alpha_{N_{c},N_{k}}^{\left(l\right)}W^{\left(l\right)}{\vec{f}}_{N_{k}}\right)$$
where σ is the sigmiod activation function, and $\alpha_{N_{c},N_{k}}^{\left(l\right)}$ is the $l^{th}$ attention coefficient after normalization, and $W^{\left(l\right)}$ is the attention weight matrix.

MGANet uses the fixed symmetric normalized Laplacian operator widely used in graph convolution as the propagation mode to dynamically learn the spatial structure through the adjacency matrix of the current channel, and introduces a multi-head attention mechanism to match and update the features of adjacent nodes of each node that has significant influence on itself. The model framework is shown in Figure 2.

#### 3.2.2. Spatial Domain Attention

The spatial attention mechanism is introduced into MGANet to automatically update the attention coefficient to extract the dynamic spatial valuable information of signals, and its definition is as follows:(5)$$A_{S}=P_{s}\cdot \sigma ((f^{\left(l-1\right)}Q_{1})Q_{2}{\left(Q_{3}f^{\left(l-1\right)}\right)}^{T}+Q_{s})$$
where $P_{s},Q_{s},Q_{1},Q_{2},Q_{3}$ is an adjustable parameter, and the attention matrix is softmax normalized, as follows:(6)$$A_{S}^{\prime }(i,j)=\mathrm{softmax}(A_{S}(i,j))$$
where $A_{S}(i,j)$ represents the correlation degree of sleep nodes *i* and *j* in the spatial domain. After the spatial attention mechanism, the spatial features ${\vec{f}}_{s}^{A}$ are updated as follows:(7)$${\vec{f}}_{s}^{A}=({\hat{F}}_{1},{\hat{F}}_{2},\cdots ,{\hat{F}}_{n})=(F_{1},F_{2},\cdots ,F_{n})A_{S}^{\prime }$$

### 3.3. Temporal Node-Level Graph Attention Sleep Network

EEG is a typical distraction time signal with intense time dependence between adjacent sleep stages. Combining the gated recurrent unit (GRU) to MGANet is used to capture temporal dependent features and spectral information of the current node, and the attention mechanism is used to capture the dynamic time information between sleep stages.

#### 3.3.1. Time-Frequency Feature Extraction

RNN represents the input epoch sequence as ${\vec{S}}_{t+1}^{N_{c}}=\{i_{t+1,1}^{N_{c}},i_{t+1,2}^{N_{c}},\ldots ,i_{t+1,n}^{N_{c}}\}$, recursively combining the time-domain features extracted at the previous moment with the current moment, thus dynamically updating the output data, that is ${\vec{f}}_{t}=r(i_{t+1,n}^{N_{c}},f_{t-1})$. Time-frequency domain features of the current node and nonlinear function combine two kinds of information sources, which are represented by ${\vec{f}}_{t}$ and r(⋅,⋅). The GRU update procedure is as follows:(8)$$\begin{array}{l} u_{t}=\sigma (W^{\left(u\right)}[{\vec{f}}_{t},i_{t,n}^{N_{c}}]+U^{\left(u\right)}[h_{t-1},i_{t+1,n}^{N_{c}}]+b^{\left(u\right)})\\ d_{t}=\sigma (W^{\left(d\right)}[{\vec{f}}_{t},i_{t,n}^{N_{c}}]+U^{\left(d\right)}[h_{t-1},i_{t+1,n}^{N_{c}}]+b^{\left(d\right)})\\ {\hat{h}}_{t}=\phi (W^{\left(h\right)}[{\vec{f}}_{t},i_{t,n}^{N_{c}}]+U^{\left(h\right)}(d_{t}⊙h_{t-1})+b^{\left(h\right)})\\ h_{t}=u_{t}⊙h_{t-1}+(1-u_{t})⊙{\hat{h}}_{t} \end{array}$$

Input sleep sequence ${\vec{S}}_{t+1}^{N_{c}}$, output updated sleep features $x_{t}$ through GRU update gate, $W^{\left(\cdot \right)}$ and $U^{\left(\cdot \right)}$ are the weight matrix of input and of the current moment ${\vec{f}}_{t}$ and the previous moment ${\vec{f}}_{t-1}$, respectively, with $b^{\left(\cdot \right)}$ representing the bias in the model. At the same time, the reset threshold vector $d_{t}$ is reset to the transition vector ${\hat{f}}_{t}$ representing the hidden state vector of sleep as the intermediary, and the time-frequency features ${\vec{f}}_{t}$ of the final output are obtained after nonlinear operations, such as sigmoid σ(⋅) and tanh ϕ(⋅). The GRU network unit structure is shown in Figure 3.

#### 3.3.2. Temporal Domain Attention

To better extract the temporal and frequency domain sleep features, the temporal attention mechanism is added to the RNN temporal sleep network, as follows:(9)$$A_{T}=P_{v}\cdot \sigma (({\left(f_{t}^{\left(r-1\right)}\right)}^{T}P_{1})P_{2}(P_{3}f_{t}^{\left(r-1\right)})+Q_{v})$$
where $P_{v},Q_{v},P_{1},P_{2},P_{3}$ are the adjustable parameters of the model, and ${\vec{f}}_{t}^{\left(r-1\right)}$ is the input feature vector after the convolution of $r^{\mathrm{th}}$ layer. The final time-domain feature expression is obtained after softmax normalization, as follows:(10)$$A_{T}^{\prime }(i,j)=\mathrm{softmax}(A_{T}(i,j))$$
where $A_{T}(i,j)$ represents the time correlation degree of sleep nodes *i* and *j*. After the time attention mechanism, the model updates to ${\vec{f}}_{t}^{A}$ which is as follows:(11)$${\vec{f}}_{t}^{A}=({\hat{F}}_{1},{\hat{F}}_{2},\cdots ,{\hat{F}}_{T_{r-1}})=(F_{1},F_{2},\cdots ,F_{T_{r-1}})A_{T}^{\prime }$$

In this paper, we present the calculation of the MGANet network layer in pseudo-code referring to Section A.3.

### 3.4. Stage-Level Attention Transitional Estimator

The immediate task by the transitional stage estimator (TSE) mainly comprises two parts: spatial graph features and temporal contextual relationships. It is challenging to classify transitional epochs because of the easily confused features of multiple stages. Therefore, we extracted the sleep features of the transition stage through EEG-learnable hidden relationships, and explicitly proposed an attention-guided transitional stage estimator.

TSE firstly aggregates spatial features and temporal features as follows:(12)$${\vec{f}}^{\prime }={\vec{f}}_{s}^{A}||{\vec{f}}_{t}^{A}$$

Then, the attention vector is represented as the features of a sleep transition stage at each epoch, and uses the softmax function to calculate the sleep stage probability distribution $P_{f^{\prime }}$ as follows:(13)$$P_{f^{\prime }}=\mathrm{softmax}(F_{C}(G({\vec{f}}^{\prime })))$$
where $F_{C}$ indicates the full connection layer used for decision-making and *G* indicates the global average pooling operation. $P_{f^{\prime }}$ represents the confidence of each sleep stage, and the class probability jointly carries the unknown stage information. As the potential feature information, the probability of the sleep stage is calculated by a series of linear or non-linear functions, such as sigmoid, to represent its attention vector as follows:(14)$$A_{f^{\prime }}=\mathrm{sigmiod}(F_{C}(P_{f^{\prime }}))$$

Finally, class probabilities and their attention vectors are clearly represented by the Hadamard product multiplication operation as follows:(15)$${\vec{f}}_{*}^{\prime }=A_{f^{\prime }}⊗P_{f^{\prime }}$$

The obtained ${\vec{f}}_{*}^{\prime }$ contains epoch features and transitional fusion information. The TSE’s network structure is shown in Figure 4.

### 3.5. Spatial Multi-Layer Graph Attention Convolution

We extend the definition of spatial graph attention convolution to nodes with multiple channels to effectively solve the problem of too large or too small weighting, caused by uneven importance distribution of adjacent nodes. After multi-head graph attention sleep feature learning, MGANet extracts spatial features ${\vec{f}}_{s}$ related to sleep; the spatial multi-layer graph architecture is shown in Figure 5.

We introduce a multi-layer graph attention convolution layer by layer using a stacked method to capture features. After extracting the center node’s features ${\vec{f}}_{c}$ and the neighbors’ features ${\vec{f}}_{k}$ in the $l^{th}$ layer, the input updates to ${\vec{f}}_{c}^{\left(l+1\right)}=\mathrm{ReLU}(W^{\left(l\right)}{\vec{f}}_{s}^{\left(l\right)})$ and the ${\left(l+1\right)}^{th}$ layer is obtained through ReLU activation. Where $W^{\left(l\right)}$ is the shared learnable weight matrix of $l^{th}$ layers. We obtained the final spatial convolution feature after the L attention layer by matrix stacking. Using a multi-layer network can improve the efficiency of model operations and reduce model parameters. The center node ${\vec{f}}_{c}^{\left(l\right)}$ of the layer l has m sleep features, which are aggregated with the attention vector $\alpha_{ij}$ for (n−l) times, combined with the weight matrix $W_{l}$ to finally obtain the center node ${\vec{f}}_{c}^{\left(n\right)}$ of layer n and its neighbor features ${\vec{f}}_{1}^{\left(n\right)}$, and the sleep features are extracted layer by layer using the stacking method.

## 4. Results and Discussion

### 4.1. Classification Performance

Our model is verified on the ISRUC [31,32] and SHHS [33,34] datasets. The obtained model performance is shown in Table 1. Dataset introduction and baseline settings refer to Section A.1 and Section A.2.

Table 1 shows the performance comparison of several baseline results. The proposed model has the best performance on the two datasets. The accuracy, MF1, and Kappa of the models were 0.825, 0.814, and 0.775 on the ISRUC dataset, and 0.873, 0.801, and 0.827 on the SHHS dataset, respectively. 

After comparing the deep learning method combining CNN and RNN, we found that the spatial location information of EEG could not be completely characterized, and the extracted features did not include regional connections between the brains. As shown in Table 1, the performance of traditional methods using CNN and RNN needs to be improved. Although the existing graph convolutional networks can fully extract the temporal context and spatial features, their spatial feature extraction does not consider the weight of each adjacent feature, resulting in some features being excessively important or neglected, resulting in differences between the extracted features and the real signal expression. Through the comparative test, it can be seen that the MSTGCN proposed can extract the spatial location information of EEG signals, and fully consider the importance of the interaction between adjacent channels to achieve the most advanced performance.

For each sleep stage, our model achieved the highest F1-score at the Wake, N1, and N2 stages, and the second highest level at the N3 and REM stages. Thus, for the most common confusion in the sleep staging task transition duration, namely Wake and N1, the proposed model’s proposed TSE achieves the most advanced level, with N1 performing 2% better than the second most advanced.

### 4.2. Number of Parameters and Training Duration

We compared the number of training parameters and training duration of several advanced models and MGANet models in each epoch in Table 2.

As can be seen from Table 2, the number of parameters of the proposed model is significantly less than that of other models, and the training duration of this model is also better than the others. Although the model establishes a complex graph channel structure, the parameter number of the proposed model is 1.5 × 10^5^, which is still on the same order of magnitude as other state-of-the-art methods. Furthermore, the number of parameters affects the complexity and robustness of the model; the experiments prove that the proposed model achieves better results under the same parameter magnitude.

### 4.3. Hypnograms

To visualize the classification performance of the proposed model, we compared the sleep classification results of the MGANet model with sleep hypnograms manually scored by sleep experts, as shown in Figure 6. The differences between MGANet and expert manual scoring were mostly in the N1 and REM stages. Although the proposed model effectively improved the classification accuracy of the N1 stage; the class imbalance of N1 stage causes the classification accuracy to drop.

Figure 6 shows that the MGANet projected for 800 epochs can correctly classify most sleep stages and has a high accuracy in each transition stage. In the first 20 epochs, the model mistakenly confused the N1 and W phases, which is consistent with the human sleep pattern, because humans usually repeatedly cycle between the waking to light sleep stages during this period. Around the 330 epoch, the model confuses REM with N2, a period of human dreaminess. Some incorrect results in this model are understandable because of the confusing nature of the transition stage, which is characterized by multiple sleep stages. Actually, sleep experts also inaccurately score these stages when manually scoring. Most of the results are a misjudgment of N2 as N1, or REM as N2, which are also widespread inconsistencies in other baseline models.

### 4.4. Ablation Experiments Results

In order to compare the performance of the proposed model in each module, we designed the following ablation experiments. The variable settings and experimental results are shown in Table 3 and Figure 7. The baseline is set as the spatial multi-layer graph attention convolution module, variable A is the RNN convolution, variable B is the spatial-temporal attention, and variable C is TSE. Table 3 shows that the performance of the model using a baseline to extract the non-Euclidean spatial features of sleep is poor, and the model accuracy is only 80.9%. After RNN is added, the model can extract time-dependent features well, and the model performance is slightly improved to 81.5%. However, when spatial attention is added to the baseline, the model performance deteriorates compared to the baseline. It could be that the first is the cause of the lack of time domain signal feature extraction and the second is why the model of the network layer number is less. 

Figure 7 shows eight confusion matrices obtained from ablation experiments, demonstrating the effectiveness of the proposed model in improving the classification of transition stages. In the past decades, many researchers have discussed the problem of stage-class imbalance [35] in sleep datasets, which seriously affects the classification performance of models. The transitional N1 stage accounts for only about 5–10% of a sleep cycle, while REM accounts for about 20–25% [36]. Dataset class imbalance tends to exacerbate the confusion degree in the transition stage. For this reason, traditional methods use oversampling and undersampling to expand or reduce the amount of data to achieve the purpose of data balance, but such methods can easily cause data overfitting [37]. In addition, some researchers have used weighted cross entropy loss function to balance the data. When the model performance is poor, the learning speed becomes fast; conversely, the learning rate decreases and the model performance cannot be further improved. Because the transitional stage of sleep confuses the characteristics of multiple stages, it makes it difficult to stage sleep. The current experiments have shown that the performance of the model is significantly improved after adding TSE, and it has also been proved to be helpful for accurate classification of confusion stages. Finally, the results show that the proposed MGANet adds variables A, B, and C at the same time, and the model has the best performance; its accuracy, F1 score, and kappa were 82.5, 81.4, and 77.5, respectively. Suboptimal results can be obtained using RNN and transition stage estimation methods.

### 4.5. Heads Numbers

We also discussed the number of heads of multi-head attention proposed, and designed it so that when the number of heads h = 2, 4, 8, 16, or 20, the accuracy and F1-score were obtained to explore the optimal number of heads used in Figure 8. h = 8 achieved the highest accuracy and F1-scores of 82.5 and 81.4, respectively.

In Figure 8, the head numbers of MGANet represent the standard for distinguishing features of center nodes. Target features are extracted from multiple angles to avoid feature redundancy or single features. As shown in Figure 8, the model achieves best performance when h = 8 is used. When the number of attention heads is small, such as h = 2 or h = 4, the accuracy of the model is only 80.3% and 80.7%, respectively. Due to the lack of aggregation features in each head, it results in ineffective multi-attention mechanisms. When the number of attention items are larger, such as h = 16 and h = 20, the performance gradually decreases, which may be caused by the increase of aggregation features leading to over-fitting of the model. The number of heads determines the influence of adjacent channels on the weighting of the current channel, and selecting an appropriate number of heads can achieve the best effect of the model.

### 4.6. Speed Training

We compared the training rates of the proposed model with STGCN [17] and MSTGCN [27] on 80 epochs. The experimental results are shown in Figure 9. The proposed model converges faster and the loss is closer to 0 than others.

As shown in Figure 9, the losses of the proposed model decreased the fastest among the first 10 epochs, reaching the lowest losses compared with the other two methods. Based on the speeding training in Figure 9, we consider that the multi-head graph attention convolution used by MGANet aggregates features, calculates the corresponding weights, and obtains the most refined features for model training. Therefore, the running time and running efficiency are better than other methods. It is worth noting that the proposed model converges faster and loses less than other methods.

### 4.7. Attention Channel Visualization

We visualized the MGANet constructed through nodes and edges in the brain, as shown in Figure 10. A graph structure was constructed of the human brain. The MGANet was used to correspond to the current node and that of its neighbors to establish the adjacent relationship, extract the dynamic spatial features, and display the schematic diagram of multi-orientation construction.

Figure 10 shows that the multiple neighbor nodes are aggregated by adjacent edges carrying information to form the feature representation of the current node. For the central node, the non-Euclidean distance and weight of different neighbor nodes are significantly different, so it is necessary to consider the description of their mutual influence. The dynamic update feature is faultlessly implemented in the MGANet network.

## 5. Conclusions

In this paper, we proposed a novel and state-of-the-art algorithm for a multi-layer spatial graph attention convolutional network for sleep staging. The proposed method uses graph attention convolution to process non-Euclidean sequences in the spatial dimension, RNN to process spectrum information and time dependence in the temporal dimension, and the attention mechanism causes the model to focus on attributing the most appropriate weightings to adjacent channels. The proposed transitional stage estimator makes the model more suitable for the sleep transition classification stage and solves the problem of poor classification accuracy caused by the confusion of transition stages. The proposed MGAN not only outperforms the advanced works, but also achieves the most advanced performance on the ISRUC-S3 and SHHS.

Most EEG research has been carried out in a task-driven manner, which would be largely limited by data labels to a great extent. It is difficult and expensive to obtain labeled datasets; hence, the proposed model cannot be well generalized and is not suitable for clinical diagnosis at present. In future research, we will build a semi-supervised model and an unsupervised model, and will use the easily obtained unlabeled data to classify sleep stages to improve generalization ability.

## Figures and Tables

**Figure 1 sensors-22-09272-f001:**
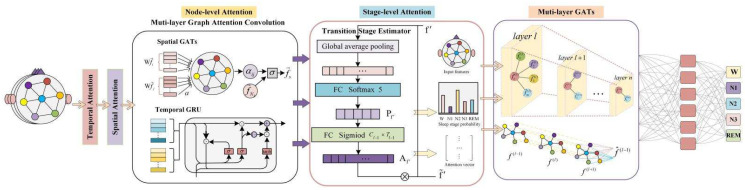
Overall architecture of the MGANet method.

**Figure 2 sensors-22-09272-f002:**
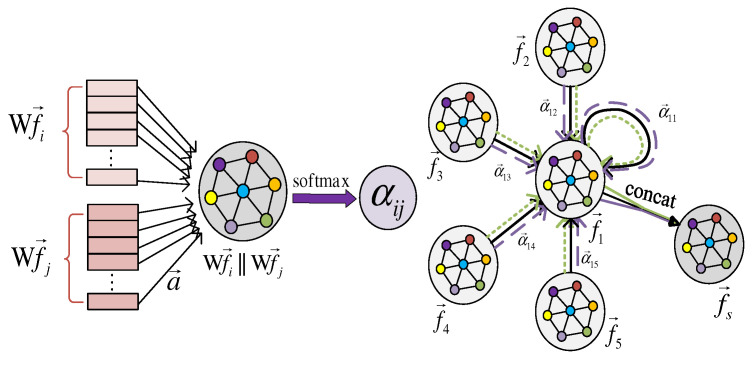
Multi-head graph attention learning structure.

**Figure 3 sensors-22-09272-f003:**
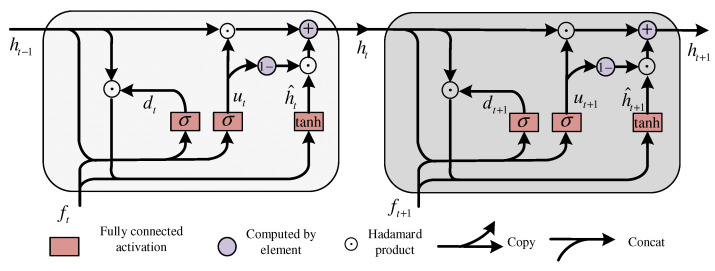
Gated recurrent unit network.

**Figure 4 sensors-22-09272-f004:**
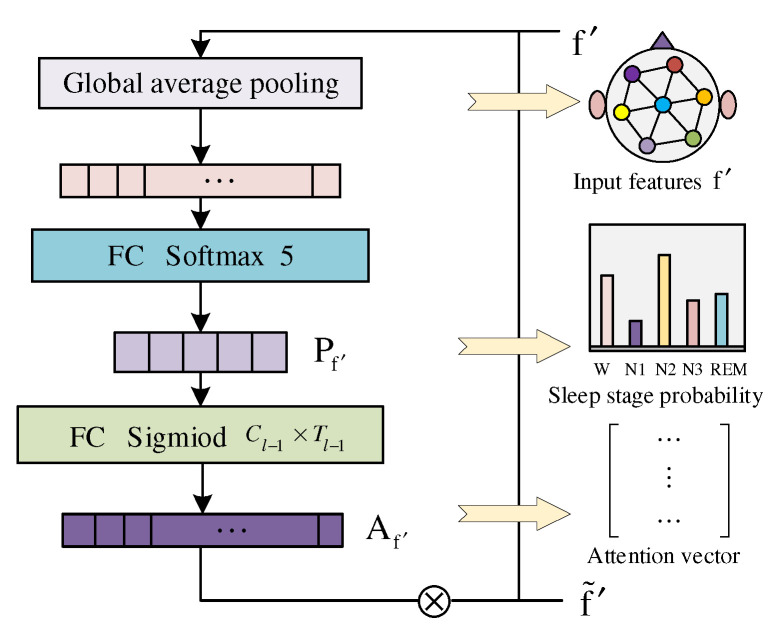
TSE consists of global average pooling, full connection, softmax, and sigmiod.

**Figure 5 sensors-22-09272-f005:**
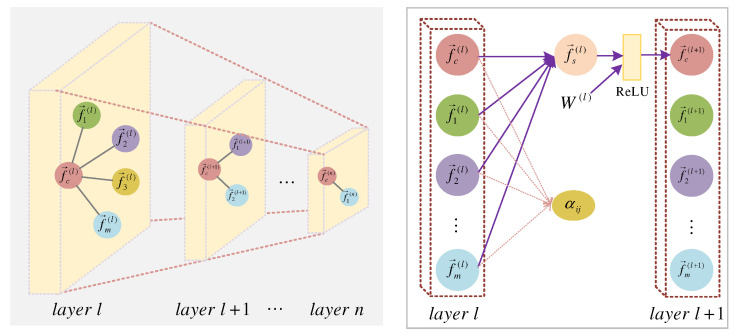
Spatial multi-layer graph attention convolution.

**Figure 6 sensors-22-09272-f006:**
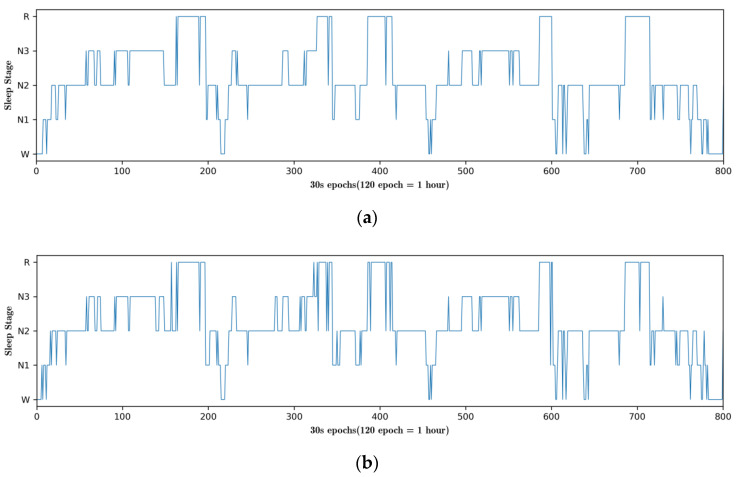
The scored hypnogram by sleep experts and proposed MGANet. (**a**) Hypnogram manually scored by sleep experts. (**b**) Hypnogram automatically scored by the proposed MGANet.

**Figure 7 sensors-22-09272-f007:**
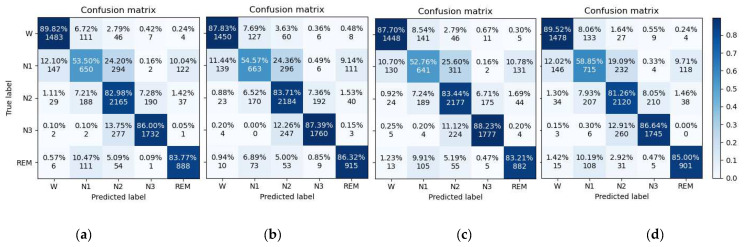
Confusion matrix obtained from ISRUC-S3 dataset for each experiment. (**a**) Baseline. (**b**) Baseline + A. (**c**) Baseline + B. (**d**) MGAN. (**e**) Baseline + A + B. (**f**) Baseline + A + C. (**g**) Baseline + B + C. (**h**) MGAN.

**Figure 8 sensors-22-09272-f008:**
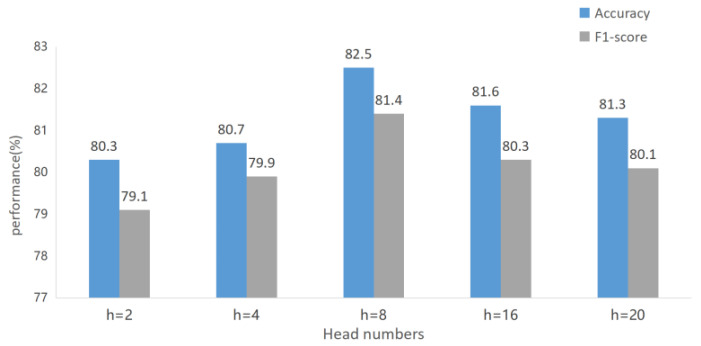
Accuracy and F1-score of the model obtained by different head numbers.

**Figure 9 sensors-22-09272-f009:**
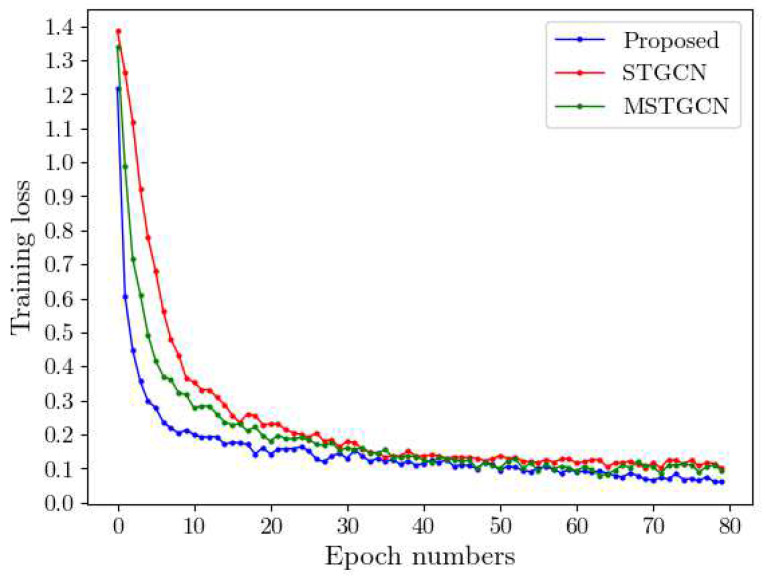
Comparison of training losses with STGCN and MSTGCN over 80 epochs.

**Figure 10 sensors-22-09272-f010:**
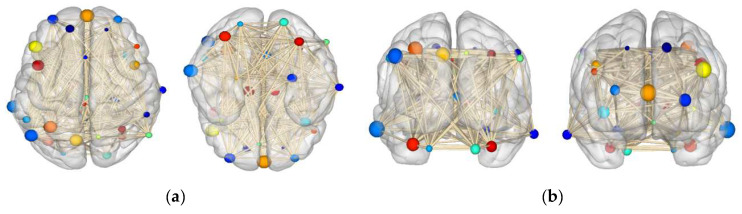
Visualization of MGANet spatial features. (**a**) Topside and bottom side. (**b**) Frontal side and backside. (**c**) Medial view of left and right. (**d**) Lateral view of left and right.

**Table 1 sensors-22-09272-t001:** Performance comparison of the state-of-the-art approaches.

Model	ISRUC			SHHS		
	Accuracy	MF1	Kappa	Accuracy	MF1	Kappa
MRCNN + AFR [25]	0.606	0.552	-	0.689	0.557	-
CNN + BiLSTM [10]	0.788	0.779	0.730	0.719	0.588	-
MLP + LSTM [12]	0.779	0.713	0.758	0.802	0.779	0.792
ResNet-50 [26]	0.782	-	0.674	0.837	-	0.754
ARNN + RNN [22]	0.789	0.763	0.725	0.865	0.785	0.811
STGCN [15]	0.821	0.808	0.769	-	-	-
proposed model	**0.825**	**0.814**	**0.775**	**0.873**	**0.801**	**0.827**

We bold the optimal result and underline the second-best result.

**Table 2 sensors-22-09272-t002:** Number of parameters and training duration for each epoch.

Model	Parameter	Training Duration
Supratak et al. [10]	1.7 × 10^7^	95.2
Dong et al. [12]	1.7 × 10^8^	268
Chambon et al. [11]	2.0 × 10^5^	9.98
Phan et al. [22]	1.2 × 10^5^	367
Chriskos et al. [23]	4.1 × 10^5^	75.8
proposed model	1.5 × 10^5^	69.7

**Table 3 sensors-22-09272-t003:** Ablation model settings.

Model Variable	Model Performance		
	Accuracy%	F1-Score%	Kappa%
Baseline	80.9	79.6	75.4
Baseline + A	81.5	80.3	76.2
Baseline + B	81.0	79.4	75.5
Baseline + C	81.4	80.3	76.1
Baseline + A + B	81.9	80.6	76.7
Baseline + A + C	82.4	81.1	77.3
Baseline + B + C	81.6	80.2	76.3
Baseline + A + B + C(MGANet)	**82.5**	**81.4**	**77.5**

We bold the optimal result and underline the second-best result.

## Data Availability

In this article, we used SHHS and ISRUC data sets. Sleep Heart Health Study (SHHS) is a multicenter cohort study of the cardiovascular and other consequences of sleep-disordered breathing. The data download link: https://sleepdata.org/datasets/shhs, accessed on 10 May 2021. ISRUC-S3 contains 126 recordings of 118 unique subjects recorded between 2009 and 2013 at the Center for Sleep Medicine, University Hospital, Coimbra, Portugal. The third panel we used contained recordings of physiological signals from 10 healthy subjects, 9 males and 1 female. The data download link is as follows: https://sleeptight.isr.uc.pt/ISRUC_Sleep/, accessed on 30 May 2021.

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
