# Peer review of "Multi-Layer Graph Attention Network for Sleep Stage Classification Based on EEG"

_sensors, 2022, doi:10.3390/s22239272_

Round 1
Reviewer 1 Report
The manuscript proposes a multi-layer graph attention network (MGANet) for sleep stage classification based on EEG signals. The method is experimentally validated and compared to the competitive approaches, showing promising results.
The manuscript is well-written and easy to follow. The main contributions of the study are stated. The proposed method is described in detail. The method is validated and compared to state-of-the-art approaches. The results are well presented.
However, here are some comments I would like the authors to address before the manuscript is considered for publication:
1. Please correct the term “non-European space” in line 50.
2. Please add a paragraph describing the manuscript’s structure at the end of the Introduction section.
3. In the literature review, please briefly state the advantages of your proposed and improved approach based on the MGANet over the recent studies utilizing conventional CNN architectures with the 2D time-frequency signal representations. Please provide some examples for these alternative approaches in various applications, as, in my opinion, your proposed method is also not limited only to application to EEG signals but could be extended to other applications in future work. Please consider briefly mentioning here the following papers for illustration purposes: 10.1007/s10044-020-00921-5, 10.1109/ACCESS.2021.3139850, 10.1109/TNNLS.2020.3008938.
4. Please improve the resolution of Figure 6.
5. Please provide a more detailed discussion of the results presented in the form of graphs and tables.
6. In the Conclusions section, please address some limitations of the presented study.
7. In the Conclusions section, please also provide some directions for future work.
Author Response
请参阅附件。

Reviewer 2 Report
The title “Multi-Layer Graph Attention Network for Sleep Stage Classification Based On EEG” is written well. However, the script has some drawbacks. Kindly include all my suggestions in the modified script for further betterment.
1. The unnecessary capitalizations should be removed as follows: Multi-Layer Graph Attention Network for Sleep Stage Classification based on EEG.
2. Concrete results of the proposed methodology should be included in the abstract.
3. Keywords should be in alphabetical
4. et al. should be in italics
5. Review of literature is not effective
6. Mention contributions in a separate section
7. There is a problem in line number 109
8. The proposed technique looks so complicated, a part of that normal deep learning techniques providing better performance in recent literature; how are authors justifying this?
9. There are problems in equations; please recertify once again
10. In my humble opinion, some structures of the manuscript are inappropriate.
11. Some formulas should be described in more detail.
12. Paper motivation can be organized with a separate section; moreover, an intuitive example is necessary to make it easy to understand.
13. Pseudo code for the proposed algorithm missing
14. The language in the paper is easy to understand. However, there are several grammar mistakes, the author should double-check the paper carefully.
15. What are the limitations of the proposed model? Please clarify them explicitly in the future.
Round 2
Reviewer 1 Report
The authors have addressed my comments.
Author Response
The addition and modification of the literature have been completed. Thank you again.
Reviewer 2 Report
The authors are incorporated all of my suggestions. Kindly add the recent below literature before publication
1. @incollection{allam2022customized,
title={Customized deep learning algorithm for drowsiness detection using single-channel EEG signal}, author={Allam, Jaya Prakash and Samantray, Saunak and Behara, Chinmaya and Kurkute, Ketan Kishor and Sinha, Vikas Kumar}, booktitle={Artificial Intelligence-Based Brain-Computer Interface}, pages={189--201}, year={2022}, publisher={Elsevier} }2. A scoping review of behavioral sleep stage classification methods for preterm infants A Bik, C Sam, ER de Groot, SSM Visser, X Wang… - Sleep Medicine, 2022 - Elsevier 3. @article{sekkal2022automatic, title={Automatic sleep stage classification: From classical machine learning methods to deep learning}, author={Sekkal, Rym Nihel and Bereksi-Reguig, Fethi and Ruiz-Fernandez, Daniel and Dib, Nabil and Sekkal, Samira}, journal={Biomedical Signal Processing and Control}, volume={77}, pages={103751}, year={2022}, publisher={Elsevier} }
Author Response

(The authors gave the same response as above.)
